Subject Area:
developmental biology/bioinformatics

Keywords:
preimplantation embryonic development, pathway and function, dynamic expression, time activation, database

Authors for correspondence:
Lei Yang
e-mail: yanglei_hmu@163.com
Yongchun Zuo
e-mail: yczuo@imu.edu.cn

†These authors contributed equally to this study.

# EmExplorer: a database for exploring time activation of gene expression in mammalian embryos

Bosu Hu[1,†], Lei Zheng[1,†], Chunshen Long[1], Mingmin Song[1], Tao Li[2], Lei Yang[3] and Yongchun Zuo[1]

[1]State Key Laboratory of Reproductive Regulation and Breeding of Grassland Livestock, College of Life Sciences, Inner Mongolia University, Hohhot 010070, People's Republic of China
[2]College of Life Sciences, Inner Mongolia Agricultural University, Hohhot 010018, People's Republic of China
[3]College of Bioinformatics Science and Technology, Harbin Medical University, Harbin 150081, People's Republic of China

LZ, 0000-0002-8531-6949; YZ, 0000-0002-6065-7835

Understanding early development offers a striking opportunity to investigate genetic disease, stem cell and assisted reproductive technology. Recent advances in high-throughput sequencing technology have led to the rising influx of omics data, which have rapidly boosted our understanding of mammalian developmental mechanisms. Here, we review the database EmExplorer (a database for exploring time activation of gene expression in mammalian embryos), which systematically organizes the genes from development-related pathways, and which we have already established and continue to update it. The current version of EmExplorer incorporates over 26 000 genes obtained from 306 functional pathways in five species. The function annotations of development-related genes were also integrated into EmExplorer. To facilitate data extraction, the database also contains the following information. (i) The dynamic expression values for each development stage are matched to the corresponding genes. (ii) A two-layer search tool which supports multi-option searching, such as by official symbol, pathway name and function annotation. The returned entries can directly link to the analysis results for the corresponding gene or pathway in the analysis module. (iii) The analysis module provides different gene comparisons at the multi-species level and functional pathway level, which shows the species specificity and stage specificity at the gene or pathway level. (iv) The analysis based on the hypergeometric distribution test reveals the enrichment of gene functions at a particular stage of one organism's pathway. (v) The browser is designed for users with ambiguous searching goals and greatly helps new users to get a general idea of the contents of the database. (vi) The experimentally validated pathways are manually curated and shown on the home page. EmExplorer will be helpful for elucidating early developmental mechanisms and exploring time activation genes. EmExplorer is freely available at http://bioinfor.imu.edu.cn/emexplorer.

## 1. Introduction

Preimplantation embryo is the initial stage of mammalian embryonic development, the process of which is rather complex. During this stage, the networks comprising various molecular interactions have a critical role during a particular stage of embryogenesis [1]. Most mammalian preimplantation embryonic developments are extremely similar and highly ordered. During the early embryo process, the single-cell zygote continues to proliferate and finally forms a blastocyst, which is composed of a certain number of cells. Mammalian preimplantation embryonic development consists of a series of important

royalsocietypublishing.org/journal/rsob    Open Biol. 9: 190054

events, such as oocyte maturation, the first cell mitosis, maternal to zygotic transition, embryonic genome activation and the cell fate decision [2–5]. Fertilization is thought to be controlled by specialized gene regulatory programmes which stringently control different gene expressions during each development stage [6–9]. The significance of the studies in the preimplantation field was previously emphasized at the molecular level, which demonstrated that the transcriptome in the early development stage plays critical roles in controlling and maintaining the cellular identity. The reprogramming implementation also relies on a deep understanding of the mRNA transcriptome [10].

With the wide use of high-throughput sequencing technology, knowledge of the expression data of early embryonic development genes has increased significantly [11]. After the first transcriptome profiling obtained on Genome Array, dynamic transcriptomes from metaphase II oocytes to blastocyst, which is the terminal stage of preimplantation embryos, were successfully tested [12]. The data obtained from this technology are described as the timing sequence expression value. The experimental data greatly help researchers to decipher the regulatory networks that contribute to morphological changes in embryo development. Data analysis has demonstrated that the major maternal to zygotic transition begins from the four-cell to eight-cell stages, during which the largest genes of the zygotic genome were activated [13]. Biologists are finding effective methods to collect and classify the larger number of genes. With a highly ordered frame we can make comparisons between diverse genes and can even make comparisons at a cross-species level [14]. Through these comparisons, researchers may find stage-specific expression genes of a particular species. Furthermore, it seems to be of great value to extract species-specific expression genes.

Hitherto a few online databases have been developed for in-depth mining of development-related genes and their dynamic expression at different stages of development. Previous studies have also suggested that analysis of the gene networks and the molecular pathways at the transcriptome level may offer an entry point to the understanding of the timing mechanism [15]. Further analysis of the transcriptome revealed several patterns of changes in the transcriptional activity during the transition through compaction and blastocyst formation [8]. There are already several existing databases, such as DevMouse [16], GED [17], DBTMEE [18], MetaImprint [19] and EMAGE [1]. The time-ordered data in these online sources vividly shows the changes in each stage. However, all the databases above have the following problems: (i) they are mainly developed for mouse embryo development, so that a database for multiple species urgently needs to be established; (ii) genes are still discrete units in these datasets as they cannot be well ordered only in the timing sequence; (iii) the specificity of gene expression is closely related to its function, however there are few published databases in this field; (iv) the current online databases have not managed to make comparisons between different species. Thus, the new challenge is to implement a highly efficient organization method between genes cross multiple species. Our research team had previously reported the timing of genome-wide activation of the Kyoto Encyclopedia of Genes and Genomes (KEGG) functional pathway in bovine preimplantation embryogenesis [20]. The results have confirmed the key functional pathways of bovine embryos, including the organelle-related functional pathway, junction pathways and receptor pathways,

and that they undergo a successive time-ordered activation [21,22]. Thus, it seemed that gathering diverse genes through different functional pathways is an efficient way to study the network between different genes. Furthermore, the functional pathways have great similarity across different mammals. So collecting, categorizing and analysing each functional pathway not only can sort diverse genes, but also seems to be a way to achieve comparisons among different species.

To our best of our knowledge, EmExplorer is the first Web-based multi-species resource focused on the temporal expression of development-related genes. The latest release of EmExplorer provides a clear molecular pathway, function annotation and dynamic expression information across different stages of preimplantation development. In addition, it offers an online analysis tool to search, browse and visualize the developmental genes with their dynamic expressions during different development stages. Users can also make expression comparisons for one specific gene or functional pathway between different species by using the online tool. EmExplorer can help researchers to understand the timing expression of specific genes in each developmental stage, which may inspire biologists and enable them to carry out further studies in cell reprograming or embryo development. This database will help with further study in assisted reproductive technology and regenerative medicine.

# 2. Material and method

## 2.1. Derivation of development-related genes

The current release of EmExplorer was designed to provide early embryonic development gene-related information regarding the pathway, basic information, function annotation and the most significant temporal expression value. As shown in figure 1, the KEGG database was chosen as the original pathway data source. It was searched with specific key words, by inputting 'embryonic development', for different species. We finally got an average of 300 relevant pathways for each organism after filtering the pathway dataset manually. The current version contains five mammals, and the detailed numbers of pathways for each species are illustrated in table 1. In total, the five mammals contain 306 functional pathways. By mining the genes under the pathways for each species, we collected 13 000+ development-related genes (include 3200+ common genes and species-specific genes) for five species, including 5098 *Homo sapiens* (human) genes, 5874 *Mus musculus* (mouse) genes, 6073 *Bos taurus* (cattle) genes, 5565 *Sus scrofa* (wild boar) genes and 4290 *Macaca mulatta* (rhesus monkey) genes. The Latin name for organisms in EmExplorer corresponds to the Latin name in the National Center for Biotechnology Information (NCBI) and KEGG databases, which are specific strains for each species. The detailed information for all genes was referenced against the information obtained from NCBI. The genes were further annotated by the corresponding terms stored in Gene Ontology. The integration of function annotations provides more comprehensive information. In addition, it may give users insights into the way a gene actually functions.

## 2.2. Manual annotation at the gene level

We chose PubMed as the original literature data source. It was searched with the keywords for different early development

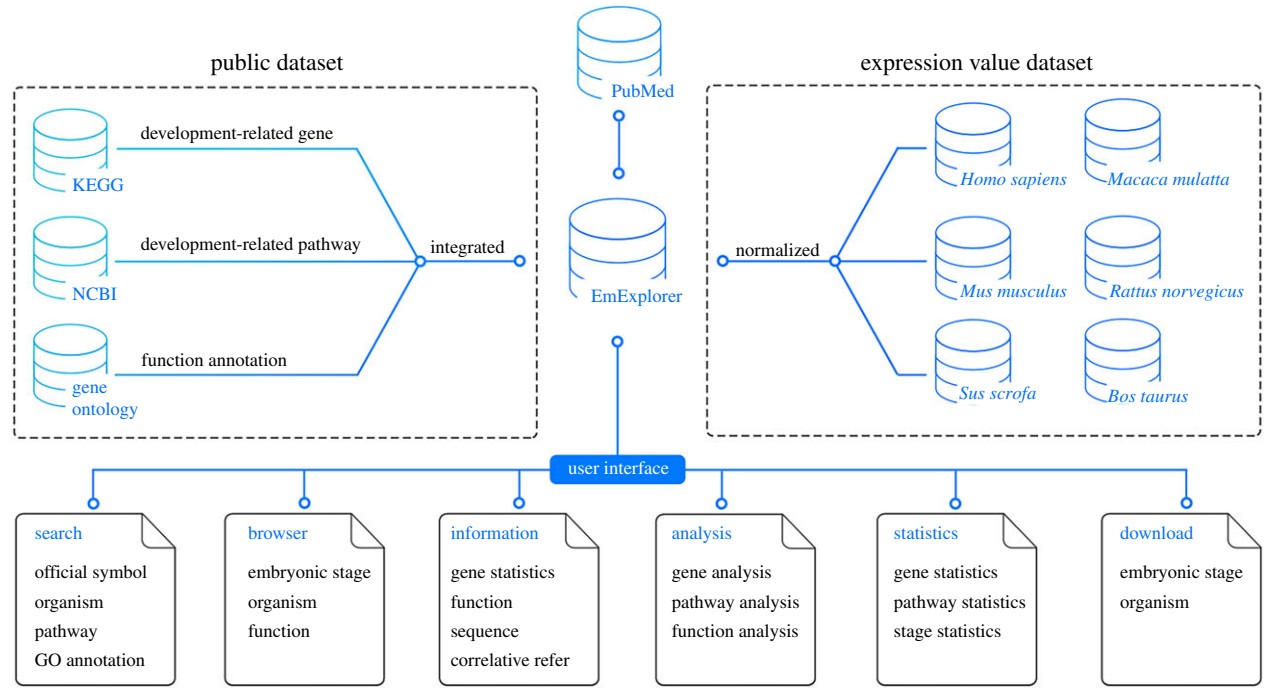

**Figure 1.** Overview of the establishment process and workflow of EmExplorer. EmExplorer integrates development-associated genes from public resources and sorts them into functional pathways. Users can input key words to the query engine, and the relevant information will be extracted from the database. The analysis tools enable users to make comparisons between genes on different levels. All search and analysis results will be helpful for further analysis.

**Table 1.** EmExplorer data content and detailed statistics .

| species | pathway | gene | function annotation | gene with dynamic expression value | gene with function annotation |
|---|---|---|---|---|---|
| Homo sapiens | 306 | 5098 | 131 | 3663 | 399 |
| Macaca mulatta | 292 | 4290 | 14 | 3123 | 24 |
| Mus musculus | 302 | 5874 | 132 | 3333 | 356 |
| Sus scrofa | 302 | 5565 | 106 | 3180 | 182 |
| Bos taurus | 302 | 6073 | 99 | 5253 | 250 |
| total | 306 | 13 040 | 195 | | 484 |

stages and species. The search returned in total more than 200 000 published articles and the relevant articles were further filtered by using keywords as follows: 'preimplantation development', 'early embryo', 'pathway', 'genome activation', etc. Finally, we found more than 17 500 relevant manuscripts as the final source, from which we extracted the popular development-related genes. From this study, we further enriched the genes associated with early development and inputted them into EmExplorer. We simultaneously counted the significance and attention of academic studies to these genes, and we took the top 50 genes as the most popular genes and incorporated them in the analysis module (mono-gene analysis tool).

## 2.3. Data processing

The dataset used in this research originates from papers published previously [12,23–25] and the experimental results from our laboratory (the expression data of pig early embryo), the sources are listed in the references and are also shown in the download module of EmExplorer. However, the expression value of *Sus scrofa* derives from the latest study in our

laboratory, but we have currently decided not to publish it. The previous work matched the gene with its pathways and used functions to annotate it. We then combined the experimental expression data with their corresponding genes.

Directly making comparisons between several genes by using the raw data in the dataset is unjustified. Take housekeeping genes as a case in point, with a low expression level a housekeeping gene can significantly influence the development process [26]. However, most genes with a much higher expression level than housekeeping genes play an insignificant role during early development [27]. In the case of using raw data, the comparison between different species is also impossible. Thus, all these expression data underwent the same normalization procedure, so that they can be compared at the same level [18]. The normalization process is simple but effective

$$\text{norm}_{\text{val}} = \frac{\text{val}_{\text{stage}}}{\sum \text{val}_s},$$

where $\text{norm}_{\text{val}}$ is the normalized expression value of a gene at a particular stage, $\text{val}_{\text{stage}}$ denotes the gene expression value at a

certain stage and $\Sigma\mathrm{val}_s$ represents the sum of the expression values of a certain gene during the entire early development process. Data processing cannot be carried out at a cross-species level. However, y using a normalizing procedure, common genes across different species can be compared at the same level. The dataset was finally stored in MySQL database.

## 2.4. Identification of a specific pathway

Before discussing figure 2, two concepts need to be clarified. First, we defined the high expression genes (HEGs) at the pathway level as being those genes whose expression value is higher than the average level of all genes in a particular pathway. We called the percentage of HEGs in all genes the high expression gene ratio (HEGr). Second, the mean value of these HEGs was abbreviated as HEGm. We assumed that the specific stage of a pathway shows high HEGr and HEGm, which means the specific stage shows its specificity in these two quantities.

We defined HEGs as having expression values higher than the mean level

$$\mathrm{HEGr} = \frac{\sum_{k \in G_p} Z_k}{\sum k \in G_p},$$

where HEGr is the percentage of HEGs in all genes in the same pathway. We denote $G_p$ as the set of genes under a particular functional pathway. $Z_k$ is a binary variable which takes the value of 1 when the gene expression level is above the median of overall genes. We further ranked expression genes in pathways according to their expression value from small to large, and then introduced HEGm, which represents the median of the HEG expression level [28]

$$\mathrm{HEGp} = \frac{3\Sigma f}{4}$$

$$\beta = \{\mathrm{HEGp}\}, \quad G_L = [\mathrm{HEGp}], \quad G_N = [\mathrm{HEGp}] + 1,$$
$$\mathrm{HEGm} = \beta \cdot \mathrm{Val}_{GN} + (1 - \beta) \cdot \mathrm{Val}_{GL},$$

where HEGp is the position of the HEG median and $f$ represents the total number of genes in a certain pathway. $\beta$ is the fractional part of HEGp. $G_L$ is the integer part of HEGp, which corresponds to the ranking number of the gene whose expression is equal to the average in the sorted expression gene list. We, respectively, defined $\mathrm{Val}_{GN}$ and $\mathrm{Val}_{GL}$ as the expression value of the GN[th] gene and the GL[th] gene in a certain development stage. Finally, we can calculate HEGm, which is the median of HEGs. The stage which has a high value of both HEGr and HEGm is denoted as the specific expression stage of a particular pathway.

We attempted to determine the specific expression developmental stage of a gene. The development stages can be divided into three categories: (i) before transition (oocyte, zygote, cell-2 and cell-4); (ii) transitioning (cell-8, called MZT); (iii) after transition (morula and blastocyst, called embryonic genome activation) [21,29]. As only a few genes are activated during the before transition stages, the embryo mostly relies on the maternal substance deposited in the oocyte cytoplasm [30]. In the after transition stage, the embryo can maintain its crucial biological process. A previous study suggested that genes that are highly expressed in this period are closely related to morphological development [31]. The obvious changes occur during the transition stage, when the majority of oogenic products are lost due to degradation [13,32]. Previous studies have indicated that the expression tendency of the functional pathways, which can be obtained by calculating the overall trend of genes in a particular pathway, also presents temporal specificity [1,33–36]. Based on the literature collected from PubMed, we finally extracted 36 experimentally validated stage-specific pathways, which are presented in the home page of EmExplorer (figure 2a), and, respectively, classified them into seven development stages (table 2). Finally, data collation, analysis of the expression value and the pathways of individual genes in the dataset were completed.

## 2.5. Functional enrichment analysis

Genes expressed specifically at a certain development stage may strongly relate to a biological function. Deeply mining the common functional effects of a group of specific expression genes may reveal that some development-related functions were enriched in a certain pathway in each stage. This part of the analysis is based on the hypergeometric test and test at the pathway level

$$p = \sum_{k=0}^{K-1} \frac{C_M^k \times C_{N-M}^{n-k}}{C_N^n},$$

where $N$ is the total number genes of a function pathway; $M$ represents the number of specific expression genes at a particular stage; and $n$ is the total number of gene functions in this pathway. $k$ is defined as the number of specific genes during a stage which has a certain function. The $p$-value was finally corrected to obtain the $q$-value. If the $q$-value of a function is less than 0.05, we could assert that these stage-specific genes of the pathway are significantly related to performing a particular biological function. The analysis results are shown in the analysis page for each pathway, as this helps users to understand what the aim of these specific expression genes is.

## 2.6. System design and implementation

EmExplorer is based on LAMP architecture: Linux operating system, Apache HTTP server, MySQL database management system, and the Web platform based on the PHP background scripting language [37]. The thinkPHP framework was used for web applications, as its strong PHP subordinate code can guarantee the stability and performance of EmExplorer Web services. The client is built on the bootstrap front-end framework to implement a user-friendly interface, and can also implement any functionality on the PC side as on the same end of a mobile device. Java Script implements visualization of statistical data in Highcharts, and users can export images in different formats. EmExplorer can be accessed at http://bioinfor.imu.edu.cn/emexplorer/.

# 3. Database use and access

## 3.1. Overview of the usage of EmExplorer

The database supports five basic operations: home, search, browser, analysis and download. As shown in figure 3, users can obtain detailed information on

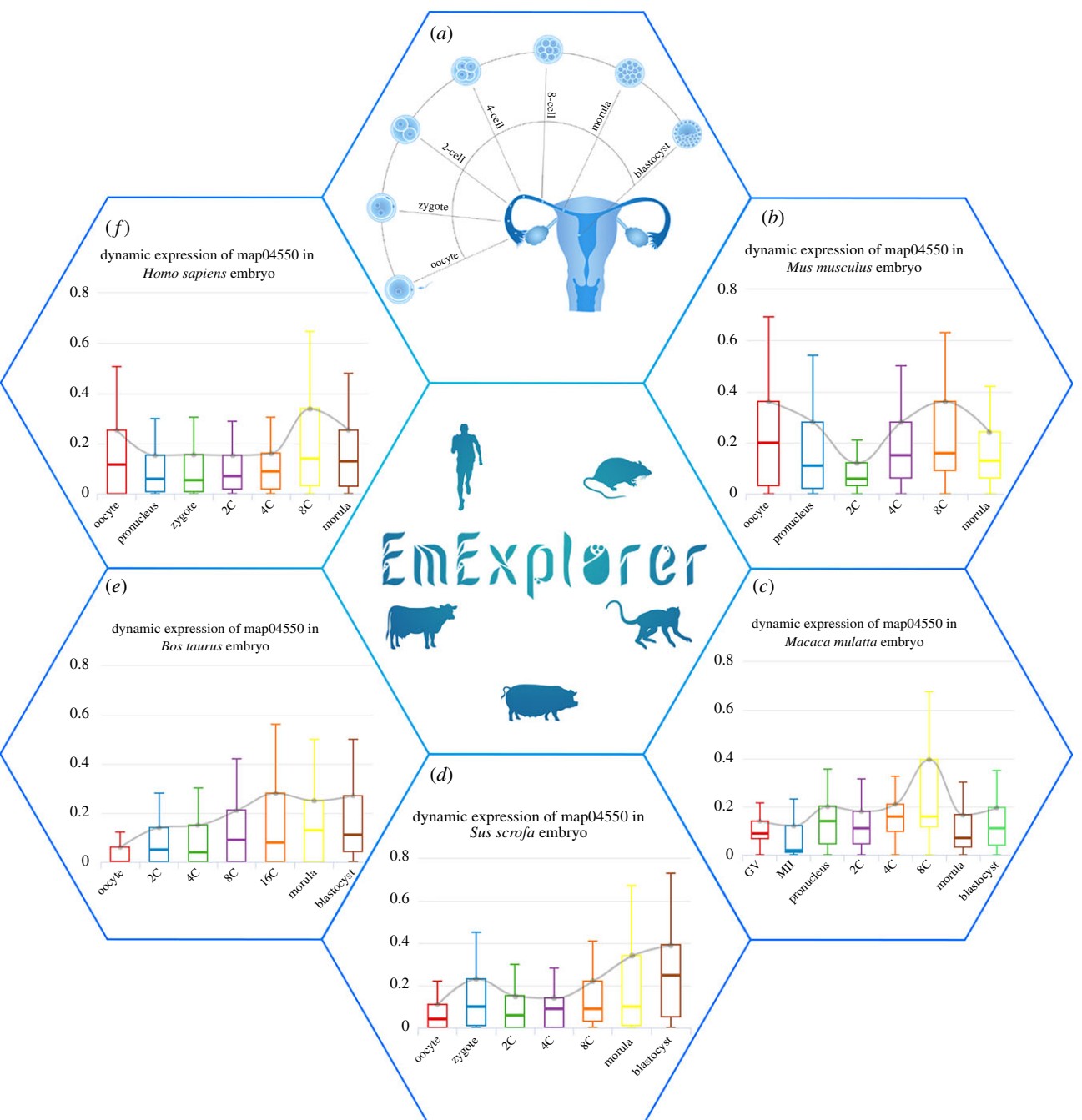

**Figure 2.** The analysis of the pluripotent stem cell (PSC) regulation pathway at the pathway level. (*a*) The experimentally validated information is provided on the home page. Each developmental stage contains several stage-specific functional pathways. We selected one functional pathway, which was confirmed by previous experiments and was the same as the results we obtained after data processing, at each stage. The current mainstream browsers can completely support the special effects of the home page. (*b*–*f*) The HEGs are defined as the group of genes whose expression values in a stage are higher than the median of the overall genes in a certain pathway. The median expression value of HEGs represents the overall level of HEGs. As the figure shows, the analysis result of the PSC regulation pathway for multiple species indicates that the common feature of these five organisms is that genes in the PSC pathway are rapidly raised at the eight-cell stage and decreased when it comes to the morula stage. The eight-cell phase may a key node for cellular pluripotency activation. Except for human and mouse, the temporal expression in the other three organisms reaches its lowest level at the oocyte stage. Such a difference between human and mouse and the other three species relates to the differences in biological functions performed in this stage.

development-related genes by exercising user-specific options. The search results are presented in a table, and users are able to click the genes in the table and be led to the website for comprehensive information. The browser was designed for new users or users who do not have a clear searching target to have a quick view of EmExplorer. Three analysis tools can be used to make comparisons at the gene level, pathway level and species level. All of the dataset hosted in EmExplorer can be downloaded to users' PC for further analysis.

## 3.2. Search

EmExplorer provides users with a convenient two-layer search tool that allows users to search directly for the records of the genes. The flexible search engine, based on a MySQL back-end, provides several query parameters, such as database ID, official symbol, pathway name, function annotation, etc. Fuzzy match was masterly applied in the module, so users need not be concerned about being off-target when they use search tool. For example, if visitors input 'IGF' in the

**Table 2.** The experimentally validated content in each developmental stage.

| stage | | *Homo sapiens* | *Macaca mulatta* | *Mus musculus* | *Bos taurus* | *Sus scrofa* |
|---|---|---|---|---|---|---|
| oocyte | pathway | 2 | 0 | 2 | 2 | 1 |
| | gene | 18 | 0 | 22 | 11 | 11 |
| | function | 21 | 0 | 22 | 14 | 16 |
| zygote | pathway | 10 | 4 | 9 | 9 | 8 |
| | gene | 77 | 3 | 81 | 55 | 42 |
| | function | 48 | 6 | 52 | 46 | 42 |
| 2-cell | pathway | 3 | 2 | 3 | 3 | 2 |
| | gene | 51 | 6 | 34 | 20 | 18 |
| | function | 53 | 7 | 43 | 38 | 38 |
| 4-cell | pathway | 15 | 6 | 15 | 15 | 15 |
| | gene | 107 | 4 | 107 | 76 | 59 |
| | function | 6 | 3 | 56 | 40 | 54 |
| 8-cell | pathway | 4 | 1 | 5 | 4 | 4 |
| | gene | 38 | 1 | 43 | 30 | 23 |
| | function | 31 | 2 | 34 | 27 | 26 |
| morula | pathway | 7 | 0 | 7 | 4 | 5 |
| | gene | 11 | 0 | 11 | 5 | 6 |
| | function | 11 | 0 | 11 | 6 | 7 |
| blastocyst | pathway | 7 | 3 | 7 | 7 | 7 |
| | gene | 74 | 2 | 64 | 55 | 31 |
| | function | 47 | 4 | 46 | 39 | 35 |

query box, the result contains 'IGF2', 'IGF2r', 'IGF3', etc., which as much as possible covers all possibilities. The quick search results are shown in figure 3; the result table summarizes the basic information of the records in the dataset which conforms to the key words. The links on these genes would bring users to the detailed information for the corresponding development-related genes. The marks on each entry link to the analysis result of the corresponding gene in the analysis module. When a query contains a lot of information, a great number of entries may return, which really frustrates users who want to quickly retrieve the information they need. In this regard, the second layer of this tool can be used for further selecting to get the most useful terms quickly. If a gene has multiple entries in the returning table, it probably means the gene has pluripotency or the gene exists in different pathways during early development.

The links for each gene have the same format as shown in figure 3, with detailed information consisting of the official symbol, genome location, functional pathway, the biological functions involved, relevant articles and the dynamic expression value ordered in timing sequence. Based on the information offered above, users can have a comprehensive view of a gene in three aspects: (i) whether the gene is located in either a genome or functional pathway; (ii) at which stage its expressing ability is active or shows its strongest expressing activity; (iii) how it influences the whole development process. Understanding the three questions above may help researchers to have a profound understanding of early embryonic development.

## 3.3. Browser

A user-friendly and configurable browser was developed to view the query results for preimplantation embryonic genes intuitively. The browser connected to the MySQL back-end allows users to select multiple species and multiple function annotations in a particular developmental stage. The result in the returning table is the intersection of the options. The information being offered by the browser, which was manually annotated by referring to over 17 500 articles, is completely credible. The browser was designed for the users who are new to the database and users whose search purpose is not clear. The browser aims to provide the most comprehensive relevant genetic information to users.

## 3.4. Analysis tool

The database also provides a series of analysis tools, each of them designed for comparison in one aspect. All these analysis tools are supported by the normalized expression values we stored in the database. At the outset, the expression levels of the same genes between different species are not in the same order of magnitude, so they cannot be directly compared. Even in the same species, there is no direct comparison between the different genes. After standardization of the data processing, all these values were eventually put on the same level for further analysis. The analysis tool consists of three independent functional parts. By using these tools, researchers can get expression information from three

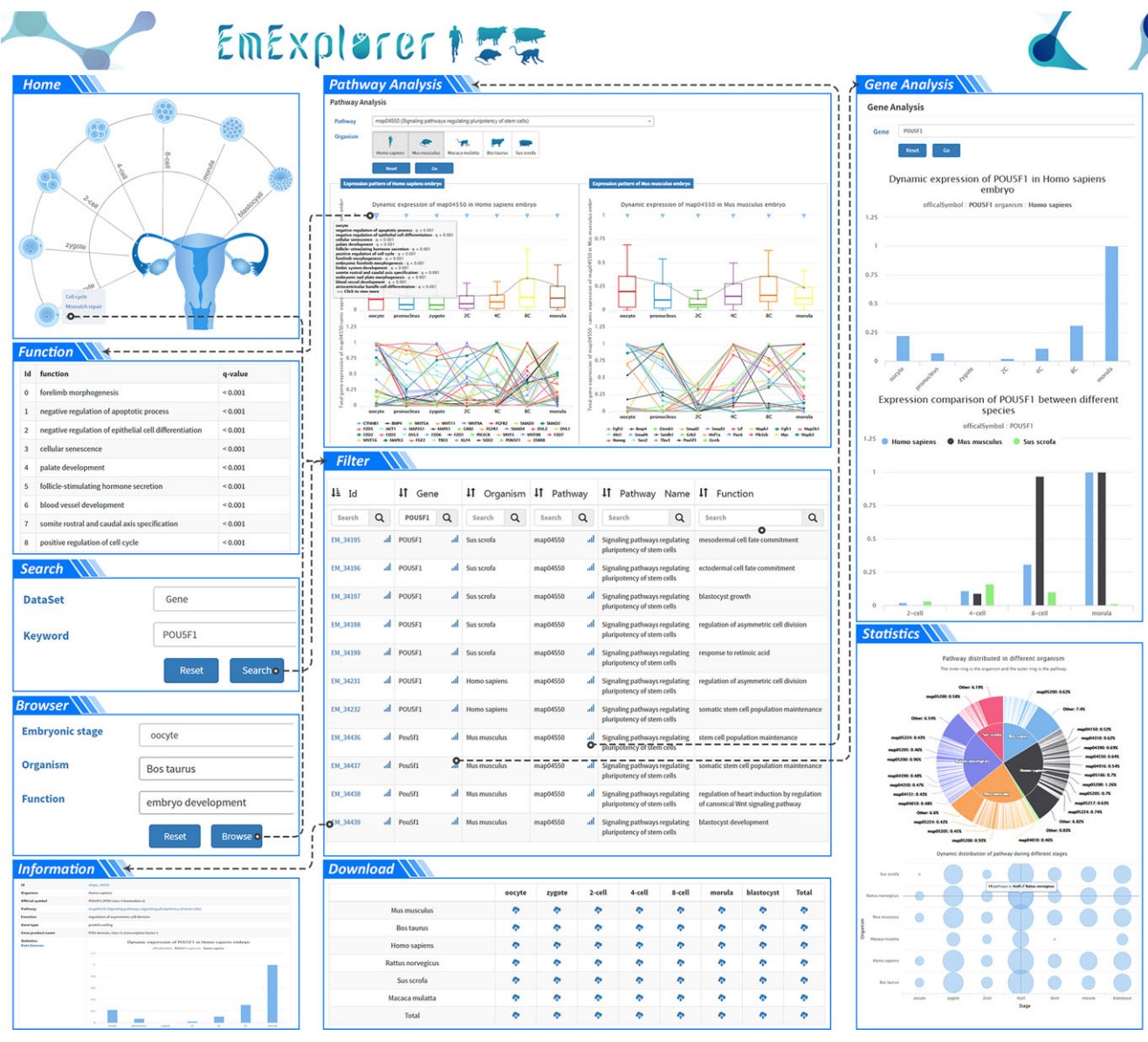

**Figure 3.** The basic operations in EmExplorer are shown above. Taking the significant gene POU5F1 as a case in point, the search browser results are presented in a table and clicking the corresponding linkage leads users to the detailed information website. Analysis tools enable users to make comparisons at the gene, organism, and pathway levels, and the results are intuitively visualized. Functional enrichment analysis shows the significance of biological functions at a certain stage. All these data can be downloaded for further analysis. The Statistics box shows the current content in EmExplorer.

aspects: at the mono-gene level, at the multi-gene level and at the pathway level. We hope this may give researchers inspiration in the following ways. Firstly, scientists can know about a stage-specific expression gene of a certain species. Secondly, the database compares dynamic expression of multiple genes in a particular functional pathway. Thirdly, the tool leads researchers to understand the overall expression level of the genes in each pathway. Researchers can have a profound understanding of a particular gene or a functional pathway through the information above. The hypergeometric test result present on each development presents the significant biological functions for an organism in a particular pathway. The series tools combine organisms, genes, pathways and functions for a comprehensive understanding of the development process.

## 4. Result

Genes stored in EmExplorer are the basic functional entries in preimplantation embryonic development. Their accurate temporal and spatial expressions play important roles in biological research of embryonic development. Over recent years, many studies have produced large-scale data to detect dynamic changes in gene expression during mammalian embryonic development. We processed, analysed and visualized FPKM changes over time during the early embryo process, and constructed EmExplorer, the first database to provide researchers with convenient access to dynamic gene expression patterns in the developmental processes of multiple species.

The analysis tools are the highlights of the database. Since the data were processed, we firstly implemented the comparison across different species. The analysis of the pluripotent stem cell (PSC) regulation pathway at the pathway level is a case in point. PSCs are cell types unique to embryonic development and are the source of all mature tissues of the organism. The temporal regulation of PSC genes is a popular research field in PSCs. As shown in figure 2b–f, the common feature of these five organisms is that the activity of the genes in the PSC pathway rapidly increases at the eight-cell stage.

Except for human and mouse, the temporal expression in the other three organisms reaches its lowest level at the oocyte stage. Through the functional enrichment analysis, the result shows that genes specifically expressed in all five organisms at the oocyte stage and enriched in functions which negatively regulate the cell cycle and morphogenesis. However, the difference between human and mouse and the other three species is that genes with positive regulation of the cell cycle are also enriched at these stages (like TBX3). The result also indicates that the eight-cell stage may indicate nodes for cellular pluripotency activation.

Furthermore, multi-gene analysis tools are used to explore the temporal expression of pluripotency factors (e.g. NANOG, POU5F1 and SOX2) in embryonic development. The result suggested that these key genes expressed highly in the morula stage may be strongly relevant to the critical roles in initiating and maintaining pluripotency. This preliminary conclusion is consistent with previous studies [24,38,39], which means that EmExplorer is able to give researchers the right insight based on existing data. Although there are similar spatio-temporal expression patterns in various species, genes with significantly different expressive features are the focus of the researchers. For example, POU5F1 is a key factor for embryonic development and stem cell pluripotency [40,41], the expression of which rapidly increases at the four-cell stage in humans and mouse. On the contrary, it smoothly expresses in other species. BMP4, a key gene that works with POU5F1 to determine cell developmental fates [39], shows high expression at the eight-cell stage in all five species; however, in contrast to the other species, BMP4 in cattle shows continuously high expression activity at the morula stage. This result supports organism specificity

in embryo development regulation. Thus, EmExplorer provides a series of tools that greatly help users to understand the similarity or difference across diverse animals.

# 5. Conclusion

The significance of early embryo research may have further clinical values. For example, the recent study suggested that, during the four-cell stage in mouse embryos, heterogeneous gene expression has a strong association with cell fate decisions, the lower expression levels of Sox21 promoting extra-embryonic fate over pluripotency [30]. The characterization of the molecular mechanism in cell fate decisions may thus bring about new types of regenerative medicine therapeutics [42]. EmExplorer provides dynamic gene expression profiles to facilitate the molecular mechanism characterization of developmental processes. By using the database, experimental researchers can investigate gene expression profiles, screen development-associated genes, make comparison between species, and download relevant data for further research.

Data accessibility. All the DNA methylation data and gene expression data were deposited at the NCBI Gene Expression Omnibus (GEO) under accession nos. GSE44183, GSE86938 and GSE59186.

Competing interests. We declare we have no competing interests.

Funding. This work was supported by the National Nature Scientific Foundation of China (grant nos. 61561036 and 61702290), Program for Young Talents of Science and Technology in Universities of Inner Mongolia Autonomous Region (NJYT-18-B01) and Fund for Excellent Young Scholars of Inner Mongolia (2017JQ04). The funders had no role in the study design, data collection and analysis, and nor in the decision to publish or in the preparation of the manuscript.

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
