## [Reviewer comments · Open Biology]

Review History

RSOB-19-0054.R0 (Original submission)

Review form: Reviewer 1

Recommendation

Accept with minor revision (please list in comments)

Are each of the following suitable for general readers?

- a) **Title**
Yes
- b) **Summary**
Yes
- c) **Introduction**
Yes

Is the length of the paper justified?

Yes

Should the paper be seen by a specialist statistical reviewer?

No

Is it clear how to make all supporting data available?

Yes

Is the supplementary material necessary; and if so is it adequate and clear?

Not Applicable

Do you have any ethical concerns with this paper?

No

Comments to the Author

The authors developed a database named EmExplorer for exploring time activation of gene expression in mammalian embryos, which incorporates over 26000 genes obtained from 306 functional pathways in 5 species. With the multiple analysis tools and visualization options, the EmExplorer database might be helpful for elucidating the early developmental mechanism, and exploring time activation genes.

Since the EmExplorer maybe the first web on temporal expression of development-related genes for mammalian embryos, it's better provide some example for key genes in EmExplorer as well as in related references, to confirm the precise of the information in EmExplorer.

Decision letter (RSOB-19-0054.R0)

03-May-2019

Dear Dr Zuo

We are pleased to inform you that your manuscript RSOB-19-0054 entitled "EmExplorer : A database for exploring time activation of gene expression in mammalian embryos" has been accepted by the Editor for publication in Open Biology. The reviewer(s) have recommended publication, but also suggest some minor revisions to your manuscript. Therefore, we invite you to respond to the reviewer's comments and revise your manuscript.

Please submit the revised version of your manuscript within 7 days. If you do not think you will be able to meet this date please let us know immediately and we can extend this deadline for you.

When submitting your revised manuscript, you will be able to respond to the comments made by

the referee(s) and upload a file "Response to Referees" in "Section 6 - File Upload". You can use this to document any changes you make to the original manuscript. In order to expedite the processing of the revised manuscript, please be as specific as possible in your response to the referee(s).

- 1) A text file of the manuscript (doc, txt, rtf or tex), including the references, tables (including captions) and figure captions. Please remove any tracked changes from the text before submission. PDF files are not an accepted format for the "Main Document".
- 2) A separate electronic file of each figure (tiff, EPS or print-quality PDF preferred). The format should be produced directly from original creation package, or original software format. Please note that PowerPoint files are not accepted.
- 3) Electronic supplementary material: this should be contained in a separate file from the main text and meet our ESM criteria (see <http://royalsocietypublishing.org/instructions-authors#question5>). All supplementary materials accompanying an accepted article will be treated as in their final form. They will be published alongside the paper on the journal website and posted on the online figshare repository. Files on figshare will be made available approximately one week before the accompanying article so that the supplementary material can be attributed a unique DOI.

Online supplementary material will also carry the title and description provided during submission, so please ensure these are accurate and informative. Note that the Royal Society will not edit or typeset supplementary material and it will be hosted as provided. Please ensure that the supplementary material includes the paper details (authors, title, journal name, article DOI). Your article DOI will be 10.1098/rsob.2016[*last 4 digits of e.g. 10.1098/rsob.20160049*].

- 4) A media summary: a short non-technical summary (up to 100 words) of the key findings/importance of your manuscript. Please try to write in simple English, avoid jargon, explain the importance of the topic, outline the main implications and describe why this topic is newsworthy.

Images

Data-Sharing

It is a condition of publication that data supporting your paper are made available. Data should be made available either in the electronic supplementary material or through an appropriate repository. Details of how to access data should be included in your paper. Please see <http://royalsocietypublishing.org/site/authors/policy.xhtml#question6> for more details.

Data accessibility section

- DNA sequences: Genbank accessions F234391-F234402
- Phylogenetic data: TreeBASE accession number S9123
- Final DNA sequence assembly uploaded as online supplemental material

- Climate data and MaxEnt input files: Dryad doi:10.5521/dryad.12311

Sincerely,

The Open Biology Team
mailto:openbiology@royalsociety.org

Reviewer(s)' Comments to Author:

Referee:

Comments to the Author(s)

The authors developed a database named EmExplorer for exploring time activation of gene expression in mammalian embryos, which incorporates over 26000 genes obtained from 306 functional pathways in 5 species. With the multiple analysis tools and visualization options, the EmExplorer database might be helpful for elucidating the early developmental mechanism, and exploring time activation genes.

Since the EmExplorer maybe the first web on temporal expression of development-related genes for mammalian embryos, it's better provide some example for key genes in EmExplorer as well as in related references, to confirm the precise of the information in EmExplorer.

Author's Response to Decision Letter for (RSOB-19-0054.R0)

See Appendix A.

Decision letter (RSOB-19-0054.R1)

14-May-2019

Dear Dr Zuo

We are pleased to inform you that your manuscript entitled "EmExplorer : A database for exploring time activation of gene expression in mammalian embryos" has been accepted by the Editor for publication in Open Biology.

Article processing charge

Please note that the article processing charge is immediately payable. A separate email will be sent out shortly to confirm the charge due. The preferred payment method is by credit card; however, other payment options are available.

Sincerely,

The Open Biology Team
mailto: openbiology@royalsociety.org

Appendix A

Dear Editor,

Thank you so much for giving us the opportunity to submit a revised manuscript (**Title: *EmExplorer: A database for exploring time activation of gene expression in mammalian embryos***) to the “**Open Biology**” for publication. We appreciate your positive comments. We also appreciate all reviewers for their constructive comments, which helped us to improve our manuscript. The detailed description response to reviewers and changes are enclosed below, with the reviewers’ comments reproduced in italics.

All references have been rearranged in the revised manuscript. With these improvements, we hope that the manuscript is suitable for publication. We believe that EmExplorer can help researchers understand the timing expression of specific genes in each developmental stage, which may give the biologists a certain inspiration and enable them to carry out further studies in cell reprogramming or embryo development.

Please let us know if there are any other questions. We would be very pleased to try our best to further improve our manuscript.

Thank you and best regards.

Yours sincerely,

Ph.D. YongchunZuo

E-mail: yczuo@imu.edu.cn

The State key Laboratory of Reproductive Regulation and Breeding of Grassland Livestock, College of life sciences, Inner Mongolia University, Hohhot, 010070, China.

Comments to the Author(s)

The authors developed a database named EmExplorer for exploring time activation of gene expression in mammalian embryos, which incorporates over 26000 genes obtained from 306 functional pathways in 5 species. With the multiple analysis tools and visualization options, the EmExplorer database might be helpful for elucidating the early developmental mechanism, and exploring time activation genes.

Since the EmExplorer maybe the first web on temporal expression of development-related genes for mammalian embryos, it's better provide some example for key genes in EmExplorer as well as in related references, to confirm the precise of the information in EmExplorer.

Response: Thank you for your positive comments. In order to verify the accuracy in EmExplorer, we selected the pluripotency factors POU5F1, SOX2, and NANOG, which are key roles in embryonic development and are hotspots of current research. In addition, we also used the BMP4 that determines cell fate together with POU5F1. Through the multi-gene analysis tool, we found consistent conclusions with the literature and observed differences in expression pattern in different species. This means EmExplorer is able to give researchers the right insight based on existing data, and greatly helping users understand the similarity or difference across diverse animals. **All of the discussion of key genes has been added in the revision MS and the necessary references have been rearranged.**